# Analysis of Food Delivery Using Big Data: Comparative Study before and after COVID-19

**DOI:** 10.3390/foods11193029

**Published:** 2022-09-30

**Authors:** Jina Jang, Eunjung Lee, Hyosun Jung

**Affiliations:** 1Nutritional Science and Food Management Department, Ewha Womans University, Seoul 03760, Korea; 2Food and Nutrition Major, School of Wellness Industry Convergence, Hankyong National University, Jungang-no, Anseong-si 17579, Gyeonggi-do, Korea; 3Center for Converging Humanities, KyungHee University, Seoul 02447, Korea

**Keywords:** food delivery, dining out trend, social media, big data, COVID-19 pandemic

## Abstract

This study examined consumers’ change in perception related to food delivery using big data before and after the COVID-19 crisis. This study identified words closely associated with the keyword “food delivery” based on big data from social media and investigated consumers’ perceptions of and needs for food delivery and related issues before and after COVID-19. Results were derived through analysis methods such as text mining analysis, Concor analysis, and sentiment analysis. The research findings can be summarized as follows: In 2019, frequently appearing dining-related words were “dining-out,” “delivery,” “famous restaurant,” “delivery food,” “foundation,” “dish,” “family order,” and “delicious.” In 2021, these words were “delivery,” “delivery food,” “famous restaurant,” “foundation,” “COVID-19,” “dish,” “order,” “application,” and “family.” The analysis results for the food delivery sentimental network based on 2019 data revealed discourses revolving around delicious, delivery food, lunch box, and Korean food. For the 2021 data, discourses revolved around delivery food, recommend, and delicious. The emotional analysis, which extracted positive and negative words from the “food delivery” search word data, demonstrated that the number of positive keywords decreased by 2.85%, while negative keywords increased at the same rate. In addition, compared to the pre-COVID-19 pandemic era, a weakening trend in positive emotions and an increasing trend in negative emotions were detected after the outbreak of the COVID-19 pandemic; sub-emotions under the positive category (e.g., good feelings, joy, interest) decreased in 2021 compared to 2019, whereas sub-emotions under the negative category (e.g., sadness, fear, pain) increased.

## 1. Introduction

“Big data” refers to information that cannot be processed with the help of traditional tools. It is a huge volume of data generated worldwide in nearly all sectors of society [1]. Big data include a large number of structured or unstructured data sets that are difficult to handle with existing management methods or analysis systems. Data are also increasing due to the spread of smart devices, the activation of social networking services (SNSs), and the expansion of the Internet of Things (IoT) [2]. The online food delivery (OFD) industry is no exception, as consumers use online platforms to order food from a variety of restaurants and have it delivered at their convenience with just a few clicks [3]. In addition, consumers share their dining experiences, emotions, and thoughts through social media, blogs, and other online platforms [4]. In the food service industry, such big data are being actively used in various fields, including product development business management, marketing, and research [1,5].

COVID-19 has had a significant impact on consumer behaviors and perceptions related to dining out. Mayasari et al. [6] reported that people were submitting fewer Google queries about “restaurant” and searching more for “delivery” and “take-away.” In the real world, both new daily COVID-19 cases and stay-at-home orders had a negative impact on restaurant consumption, especially for full-service establishments [7]. In particular, a recent study [8] reported a sharp increase in daily food delivery orders along with growing COVID-19 cases and daily death reports. As such, due to the recent rapid change in the food service market, an in-depth study of consumer perceptions in the post-COVID-19 pandemic period is very important for market recovery [9].

This study presents an empirical study on post-pandemic behaviors as the number of meals at home has increased and the purchase of delivered food, rather than dining out, has also increased. As food delivery services now account for a large proportion of dining out experiences, research related to delivery services is becoming more active. However, more multifaceted research on consumers is needed, and in particular, research on how consumers’ perceptions have changed due to COVID-19 is still insufficient. COVID-19 has become a major turning point changing the demand and service for delivery services, making it important to study changes in consumers’ perception before and after the emergence of the pandemic. Therefore, this study aims to identify consumers’ perceptions of food delivery before and after COVID-19 through big data analysis using social networks. This paper is organized as follows. The section on related studies deals with recent study trends of dining out and food delivery in the era of COVID-19 and big data analysis in the food service industry, while the third section is dedicated to the presentation of the data and methodology. The section on results highlights the most remarkable findings, and the paper ends with an exploration of critical implications, limitations, and future studies.

## 2. Related Studies

### 2.1. Changes in Dining out Patterns after the COVID-19 Pandemic

As COVID-19 spread throughout the world in 2020, quarantine policies such as social distancing and controlling private gatherings were implemented in Korea to control the spread of the virus; in the case of the United States, strong quarantine measures and the suspension of dining out services were mandatory [10]. As a result, COVID-19 has brought about serious and profound changes in our lives, with countries being blocked and citizens staying at home and being restricted from going out [11]. This situation has also had a big impact on the global food service market and, accordingly, studies related to dining out about COVID-19 are being actively conducted. As a big change compared to previous studies of COVID-19, current studies are dealing with risk perception in the context of COVID-19 in the study of consumers’ dining out behaviors [12,13,14] and how COVID-19 has affected consumers’ eating out behaviors. Research on risk perception suggests that, when catastrophic events such as a global pandemic occur, people are initially likely to be influenced by risk perception along with optimistic bias [15]. The risk that consumers perceive from dining out during the COVID-19 pandemic may be due to potential contact with the virus when dining out [12]. Byrd et al. [12] assessed American consumers’ perceptions about the risks of contracting COVID-19 from various types of restaurant foods and restaurant service types as well as their packaging. Consumers were found to be most concerned about “food served in restaurants” as a risk of COVID-19, while “food in general” was the lowest concern. In addition, “food delivered by delivery service” and “food delivery by restaurants” were ranked in the middle but were perceived to be more dangerous than “food from carryout/curbside pick-up/drive-through” and “hot/cooked food from restaurants.” Yost and Cheng [13] suggested that, in the COVID-19 era, consumers’ dining out behaviors were based on an assessment of risk perception regarding health and safety and that trust, loyalty, and transparency, which are motivators of dining out, play an important role in managing these risk-based decisions. Therefore, this study argued that the factors of transparency, trust, and loyalty should be considered as major coordination factors when evaluating customers’ motivation to dine out in restaurants during the pandemic. Meanwhile, Dedeoglu and Bogan [14] investigated how consumers’ intention to visit luxury restaurants during the COVID-19 pandemic was influenced by dining out motivation and also whether COVID-19 risk perception and trust in government play a mediating role in their relationship. The results revealed that two motivations, sociability and affect regulation, had a positive effect on luxury restaurant visit intention and that consumers’ perceptions of COVID-19 risk and trust in the government mediated the relationship between these motivational factors and visit intention.

### 2.2. Food Delivery Service in the COVID-19 Pandemic 

As the dining out business was hit hard by the COVID-19 pandemic, it was essential to switch to non-contact services to make up for the loss [10]. In addition, non-contact dining has grown rapidly due to consumers’ high demands in the COVID-19 context [15]. Among them, food delivery service has accelerated development through delivery apps or online delivery due to the development of IT technology. In particular, due to the recent COVID-19 pandemic, the need for non-contact services has led to a large portion of food service sales [16]. According to the “2021 Domestic and International Dining Trend Report,” [16] which examined the differences in the proportion of sales by restaurant operation type before and after the pandemic in Korea, store sales decreased by 16.6% on average compared to before the pandemic, while both delivery and delivery orders increased by 31.2%. Considering the proportion by method, the use of delivery apps (58.9%) was the largest. Accordingly, since the COVID-19 outbreak, research into online food delivery among food-service-related companies has been actively conducted. According to Shroff, Shah, and Gajjar [17], 368 papers using keywords related to online food delivery were included in the Web of Science core collection between 2015 and 2021. Related review papers are also appearing one after another [17,18]. Kim, Kim, and Wang [8] examined pandemic-induced changes in the demand for restaurant delivery services and analyzed the dining out delivery sales to determine consumers’ cue utilization in their delivery choices. They found that dining out delivery sales increased significantly when the pandemic spread rapidly and remained flat even when the pandemic was suppressed. Dsouza and Sharma [19] re-examined consumer perceptions after the COVID-19 pandemic and compared whether previously judged parameters had the same effect or fundamental changes. They found that consumers considered safety to be a top priority when it comes to food; furthermore, awareness of this is increasing.

### 2.3. Big Data Analysis in the Food Service Industry

With the development of information and communication technology, along with the accumulation of data and the technology to analyze it, big data analyses have become essential for obtaining valuable insights into data [1]. Accordingly, research using big data is being actively conducted not only in industry but also in academia. Big data analysis is the process of finding hidden patterns and unknown information by effectively analyzing and processing this large number of data that cannot be processed with existing databases [20]. Through this, it is used to predict the future, find optimal countermeasures, and create new values [21]. Big data analysis methods can implement estimation modeling such as linear and artificial neural networks and support vector machines through structured data [22,23]. Through unstructured data, text mining, semantic network analysis, discourse analysis, sentiment analysis, and topic modeling can identify customer needs [21,24].

This study attempted to identify consumers’ perceptions of food delivery before and after the onset of the COVID-19 pandemic by collecting and analyzing atypical data related to food delivery through social media. For this purpose, text mining, CONCOR analysis, and sentiment analysis were used. Text mining, a method of deriving meaningful keywords from unstructured data, is used for the purpose of deriving new information and knowledge [21,25]. In other words, it collects data related to natural language processing rather than structured data, establishes an analysis unit such as sentences or keywords in the text, and derives meaningful information based on the algorithm [21]. A CONCOR analysis is a method for explaining the meaning and characteristics of a cluster by classifying structural equivalence based on the correlation between keywords based on a semantic network analysis and then forming clusters with similar properties between keywords [26,27,28,29,30,31]. Finally, a sentiment analysis, also called opinion mining, is a natural language processing technique that allows researchers to automatically check the evaluations of and opinions about objects such as products and services [32]. In general, a sentiment analysis is used to classify emotions expressed in texts or convert them into objective numerical information; in a narrow sense, it can be seen as classifying positive and negative emotions in text [33]. It includes not only simply classifying positive and negative factors, but also analyzing the intention or position of the author by extracting positive and negative words [34]. Jeong and Choi [35] studied how COVID-19 affected the characteristics of using a mobile delivery order platform by using a text mining analysis to review the usage of the food service delivery order platform before and after the outbreak of COVID-19. To this end, Korean delivery application review data from 2019 to 2020 were collected and analyzed. Shin and Lee [5] conducted topic modeling and a semantic network analysis using Twitter’s big data with the keyword “Corona dining” and studied consumers’ perceptions of dining out during the pandemic era. Meanwhile, various studies related to COVID-19 have been attempted in the food-service-related field, but methods such as surveys, interviews, and secondary data analyses have mainly been applied, creating methodological limitations [5]. However, a big data analysis can be free of such limitations that make it difficult to generalize research results due to measurement errors of surveys and interviews, researchers’ subjective judgments, and limitations of samples [36,37,38]. A big data analysis is deemed useful for research in the field of food service as it is much more advantageous to obtain generalized information from a large number of data. Therefore, the current study used a big data analysis to examine consumers’ changing perceptions of the food delivery service before and after the onset of COVID-19, which can be widely used as basic data for research on dining out related to COVID-19 using big data.

## 3. Research Methodology

### 3.1. Data and Summary Statistics

The purpose of this study was to derive keywords related to “food delivery” from social media big data, as well as identify any keyword changes associated with delivery before and after the COVID-19 pandemic, to determine practical implications for the development of food service after the COVID-19 situation. To this end, relevant texts were collected from online cafes, news media, and blogs on online portal sites. Since it would have been difficult to collect data from undisclosed accounts on Facebook or Instagram, the data collection for this study was limited to blogs and cafes on Naver and Daum. These are the country’s major portal sites, which gather news data from numerous media outlets and show very active community and cafe activities; therefore, they contain the largest numbers of relevant data in Korea. So, it was considered very useful to identify consumers’ interests, current issues, and perceptions via cafes and blogs from the two portals. The data used in this study were collected from existing data using the keyword “food delivery” for a one-year period before the COVID-19 outbreak (from January 1 to 31 December 2019) and for a one-year period after the COVID-19 outbreak (from January 1 to 31 December 2021).

### 3.2. Methodology

For this study, data collected from online social media platforms were refined to examine any changes in consumer perceptions about food delivery before and after the COVID-19 outbreak. The study’s keyword for data retrieval was selected by domain experts in consideration of the purpose of the data analysis and the relevance of the retrieved keywords, and the data were collected by “The IMC,” a company specializing in big data. TEXTOM, a big data analysis solution, was employed for data extraction and analysis. The results of the frequency of important keywords extracted via TEXTOM were divided into similar groups, and then the analysis tool Ucinet6 was used to analyze meaningful relationships among the structures of connections among the keywords. Useful words were extracted, and major text-mining indexes, such as the frequency of occurrence and term frequency–inverse document frequency (TF-IDF; term frequency–inverse document frequency), were calculated through text mining. In addition, convergence of iteration correlation (CONCOR) analysis was performed to determine the associations among co-occurring words, and the analysis results were visualized. Finally, sentiment analysis was performed to identify the associations among the positive and negative emotions connected with keywords related to food delivery.

## 4. Results

### 4.1. Content Analysis

As a result of searching data with “food delivery” as a keyword, a total of 39,144 keywords were retrieved for 2019 (before the COVID-19 outbreak), and a total of 39,240 keywords were retrieved for 2021 (after the COVID-19 outbreak) (see Table 1). Given that 1000 words per channel are reported to be appropriate for TEXTOM-based data collection, the keywords collected for this study are considered sufficient. Narrative coding for food delivery involved clustering by food, sentiment, and demand/purpose (Table 2).

### 4.2. Text-Mining Analysis

According to the results of a frequency analysis (Table 3) on keywords in documents extracted using the keyword “food delivery,” “foodservice” was found to be the most commonly appearing keyword, followed by “delivery,” “famous restaurant,” “delivery food,” “foundation,” “dish,” “family,” “order,” and “delicious” in order of frequency. This result indicates how often these words appear from a search using the keyword “food delivery,” revealing their importance in such texts. Additionally, the TF-IDF values of keywords such as “foundation,” “side dish,” “pizza,” “tteokbokki,” and “sushi” were substantially higher than those of other keywords, which signifies that these keywords have a very good scarcity value within documents retrieved with the keyword “food delivery” and are significant keywords although they do not appear frequently. Since TF-IDF plays an important role in a short-term trend analysis by factoring in both the frequency of words and the interdocument irregularity of the appearance of words, it could be assumed that these keywords acted as major factors in food delivery trends in 2019. The results for the year 2021 are presented in Table 4. As in 2019, the word “dining-out” appeared most frequently, followed by “delivery,” “delivery food,” “famous restaurant,” “COVID-19,” “dish,” “order,” “application,” and “family” in order of frequency. In addition, the TF-IDF value was high for keywords such as “famous restaurant,” “discount,” “foundation,” “Delivery App.,” “pigs’ feet,” “side dish,” and “kitchen,” which means the frequency of these words in documents related to food delivery was good. This also shows that keywords such as “delivery food,” “discount,” and “lunch box,” which were not present prior to the COVID-19 outbreak, became very influential.

### 4.3. Sentimental Network Analysis

Based on the analysis results from text mining, food-delivery-related keywords were analyzed by dividing them into sentiment network and demand (purpose) network indexes. The position and role of each individual node were analyzed using semantic network indexes, and the associated attributes of highly relevant words were identified. There are four factors to consider in these indexes. First, a given variable’s higher value of degree centrality leads to a corresponding closer relationship between the variable and other variables, and thus these indexes can be interpreted as a factor that directly affects consumer sentiment and purpose. Second, a variable’s higher value of betweenness centrality means that this variable plays a greater mediating role when other variables appear. Therefore, this variable can be considered a factor that is highly dependent on consumer perceptions of sentiment and purpose. Third, a variable’s higher value of closeness centrality may indicate that when this variable interacts with other variables, a synergy effect can be created in terms of sentiment or purpose. Finally, a variable’s higher value for page rank means that this variable is more popular among consumers than other variables in terms of emotion or purpose. Therefore, this study performed a semantic network analysis of food delivery, combining the sentiments and demands (purposes) in the 2019 and 2021 data.

Table 5 presents the results of the semantic network analysis on food delivery and consumer sentiment in 2019. Regarding consumer sentiment about food delivery, the study confirmed the formation of discourses on keywords such as “delicious,” “recommend,” “delivery food,” “side dish,” “lunch box,” “popularity,” “love,” “Korean food,” and “happy” based on the degree of connection, betweenness centrality, and page link values. After each word’s group was created by clustering, a sentiment network was visualized (as shown in Figure 1). Based on the analysis results, food delivery was visualized into four categories: delicious, delivery food, lunch box, and Korean food. Accordingly, in 2019, people searched food delivery to get recommendations for places where they could order delicious and convenient delivery food.

Table 6 presents the analysis results for a semantic network linking food delivery and sentiment in 2021, which shows the formation of discourses centered around keywords such as “delivery food,” “home meal,” “recommend,” “health,” “delicious,” and “hard.” The results of the visualization were divided into three categories: delivery food, recommend, and delicious (Figure 2). In 2021 (post-COVID-19), people engaged in more food delivery searches related to recommendations than in 2019 and showed an increased perception of struggling and being worried amid a hard situation. Notably, in 2019 (pre-COVID-19), sentiment-related keywords in food delivery formed the first-ranked group, whereas in 2021, keywords related to food itself were in the top ranks, indicating that the pattern of visualized figures itself has changed rapidly following the COVID-19 outbreak.

Table 7 presents the results of a semantic network connecting food delivery and demand (purpose) in 2019. Regarding the demand for food delivery, the study confirmed the formation of discourses with keywords such as “delivery food,” “pizza,” “chicken,” “menu,” “take-out,” “food service,” “reviews,” and “Delivery App.” In particular, it was necessary to pay attention to keywords such as “pizza” and “chicken” as delivery foods showed high values in the demand analysis results. This result implies that the most important purpose of searches related to food delivery before the COVID-19 outbreak was to order universal and representative delivery foods, such as pizza and chicken. As a result of the demand network visualization (Figure 3), food delivery in 2019 was divided into three categories: delivery food, delivery, and food service. In particular, consumers performed searches to fulfill the strong demand for deliverable foods. The analysis results for a semantic network combining consumer demands (purposes) for 2021 are shown in Table 8. Based on the demands for food delivery in 2021, the study confirmed the formation of discourses on keywords such as “delivery food,” “cafe,” “home meal,” “reviews,” “order,” “lunch box,” “delivery,” “foodservice,” “COVID-19,” and “application.” Notably, unlike in 2019, the most important purpose of keywords related to food delivery in 2021 (after the COVID-19 outbreak) was not the ordering of delivery foods that consumers had always ordered, such as chicken and pizza, but the ordering of everyday foods, such as home meals, meal kits, and lunch boxes. This was also observed in the demand network visualization (Figure 4), which was divided into four categories: delivery food, reviews, COVID-19, and application. This result suggests that during the COVID-19 situation, there was an increase in consumers ordering and purchasing their daily foods using apps. This indicates that the purpose of food delivery after the COVID-19 outbreak exhibits a very different pattern, revolving around ordering everyday food items, when compared to the purpose in 2019.

The analysis results discussed thus far can be summarized as follows: The sentiment network for food delivery in 2019 was formed around discourses with keywords such as “delicious,” “recommend,” “delivery food,” “side dish,” “lunch box,” “popularity,” and “love.” In 2021, discourses with terms such as delivery food, home meal, recommend, health, delicious, and hard emerged, and it is noteworthy that the feeling that things were “hard” appeared for the first time under the influence of COVID-19.

Moreover, the demand network for 2019 showed the co-appearance of keywords such as “delivery food,” “pizza,” “chicken,” “menu,” “take-out,” and “foodservice.” In 2021, keywords such as “delivery food,” “cafe,” “home meal,” “reviews,” “order,” and “lunch box” appeared together. This illustrates a strong contrast between the purposes of ordering food before and after COVID-19.

### 4.4. Sentiment Analysis

In this study, sentiment analysis—a natural language processing method for analyzing subjective data, such as people’s attitudes, opinions, and tendencies, from texts—was employed to extract and analyze positive and negative words from the data (Table 9). For this analysis, words were first classified using a dictionary for emotional vocabulary, which was produced independently by TEXTOM, and then the frequency of words and the intensity of their emotions were calculated. In a comparison of 2019 and 2021 in relation to food delivery, the number of positive keywords among the sentiment-based words decreased by 2.85%, whereas the number of negative keywords increased by 2.85% (Table 10 and Table 11). In particular, sub-emotions under the positive category (e.g., good feelings, joy, interest) decreased in 2021 compared to 2019, whereas sub-emotions under the negative category (e.g., sadness, fear, pain) increased.

## 5. Discussion and Implications

This study identified words closely related to the keyword “food delivery” based on big data from social media and investigated consumers’ perceptions of food delivery and related issues before and after COVID-19. The study findings can be summarized as follows. In 2019, a total of 17,728 food-delivery-related keywords appeared on social media, compared to 22,000 in 2021. In 2019, frequently appearing food-delivery-related words were “delivery,” “famous restaurant,” “delivery food,” “foundation,” “dish,” “family order,” and “delicious.” In 2021, they were “foundation,” “side dish,” “pizza,” “tteokbokki,” and “sushi.” Compared to 2019, famous restaurant, discount, foundation, and Delivery App produced high TF-IDF values in 2021, indicating consumers’ changing perceptions of food delivery after the COVID-19 outbreak. These findings are partially consistent with Jeong et al.’s study [39], which found that the number of food deliveries increased drastically after the coronavirus outbreak. Jung et al.’s study [40], which explored consumers’ perceptions of dining out before and after the onset of COVID-19 using big data analysis of social media, reported that delivery emerged as one of the new topics after COVID-19. In this study, the fact that “foundation” was a keyword both before and after the coronavirus proves the high demand in the delivery market (Table 3 and Table 4). In fact, according to a study by Chang, Park, and Kang [41], the number of food service establishments in Korea continued to increase during the COVID-19 period, including the acceleration of startups based on delivery platforms. Globally, online food delivery’s worldwide sales are expected to increase from USD 296 billion in 2021 to USD 466 billion by 2027 [42]. Accordingly, online food delivery apps are benefiting greatly. In Korea, the Delivery App.—the main keyword in 2021—has the largest market share [43]. However, given that a few apps dominate the market, service complaints such as high delivery app fees, delivery costs, and long waiting times are emerging as a hot topic in food delivery service [44]. Adak et al.’s study [45] also reported that common complaints in consumer reviews of food delivery services were related to waiting time, service and food quality, and cost. Jeong and Choi [35] analyzed the main topics of delivery order platform services before and after the outbreak of COVID-19 using text-mining techniques; the three topics classified as important after the emergence of the coronavirus included “dissatisfaction with the delivery platform in the corona situation” and “the feeling of betrayal about the monopoly of the delivery order platform.”

The sentiment analysis in this study demonstrated that positive keywords decreased by 2.85% while negative keywords increased at the same rate. Consumers’ dissatisfaction with the delivery service, as previously mentioned, is believed to have provoked negative emotions among consumers, and emotional keywords identified in this study, such as “after a long time” and “expensive,” can also be interpreted in the same context. Meanwhile, in the case of fear and pain among the negative emotion categories, the frequency percentage more than doubled after COVID-19, indicating that it was significantly strengthened compared to other emotions (Table 10 and Table 11). This finding is presumed to be related to the perception of risk related to dining out as well as negative emotions about such risk in the COVID-19 context. As the previously discussed related studies indicated, consumers’ dining out behaviors are based on risk perceptions for health and safety. In their study analyzing the importance and satisfaction of selection attributes for customers when choosing a restaurant to dine out, Ahn and Cho [46] reported that a “corona risk prevention strategy” was identified among restaurant attributes that customers considered important. In addition, Dsouza and Sharma [19] noted that, after the COVID-19 pandemic, consumers became more aware of safety in their dining experience as a top priority. In particular, the demand for delivery services surged due to the preference for non-contact consumption in the wake of COVID-19, although consumers still experienced some degree of risk perception of coronavirus contact even with delivered food or packages, as shown by Byrd et al.’s study [12]. From another point of view, one of the other problems associated with delivery services that can cause negative emotions is the excessive use of disposables in packaging. The amount of waste from hard-to-recycle straws, disposable tableware, and plastic bags is rapidly increasing with the increase in the consumption of delivered food [47]. Compared with the pre-COVID-19 era, the amount of plastic waste used has risen sharply, from 776 tons in 2019 to 923 tons in 2020—more than a 18.9% increase [48,49]. Eco-guilt refers to the guilt that individuals experience when they do not take an optional action to protect the environment when making a choice with value to the environment [50]. Kwak et al. [48] analyzed the effects of environmental awareness and guilt on eco-friendly behavior in the food delivery context and found that eco-guilt has a significant effect on eco-friendly behavior as well as when regulating the relationship between environmental awareness and eco-friendly behavior. The emotion-related keywords in the current study did not include the word “guilty.” However, considering the recent important issues regarding the delivery service [51] and the importance of Kwak et al.’s findings [48], it is assumed that some negative keywords in this study are related to eco-guilt.

Meanwhile, the keywords shown in the sentiment analysis of this study are related to important trends concerning food delivery. Recently, as a mega trend related to dining out in Korea, the importance of cost-effectiveness and single-person meals due to the increase in single-person households has been cited [16]. This trend is evident from the use of the keywords “eating alone” and “cost-effectiveness” in the results of the current study (Table 5). Furthermore, the demand for late-night meals has recently increased, as common keywords identified in this study demonstrate that, both before and after COVID-19, consumers used delivery services for late-night meals (Table 5 and Table 6). In fact, according to a recent newspaper article [52], the proportion of night delivery among all delivery orders also increased from about 7.2% in 2019 to 12.2% in 2020. This study also found keywords related to the demand network, such as “tripe,” “belly,” “pigs’ feet,” and “chicken,” which are generally preferred by Koreans as late-night snacks paired with alcohol (Table 7 and Table 8).

Meanwhile, in 2021, the keywords “meal-kit” and “premium” were newly emerged; they were not present in 2019. This finding highlights trends in the delivery market, which is pursuing differentiation through the recent high-end products as well as the diversification of products [51]. In fact, major food service delivery apps in Korea recently started selling meal kits, and delivery food service providers are trying to offer premium food and services to meet the needs of consumers oriented to premium food consumption [53,54]. Accordingly, existing fast-food-centered delivery food, including chicken and pizza, is gradually being changed into a variety of healthy and enjoyable menus that can be eaten like everyday food; this phenomenon is further discussed in the results section. These results show that big data analysis methods go beyond the limitations of surveys that have no choice but to identify the fragmentary attitudes and behaviors of consumers through structured questionnaires; they can also identify embedded consumer opinions and trends that cannot be measured with existing survey methods [5,36,37,38].

Food delivery services have increased sharply since the emergence of the COVID-19 pandemic, but they are not expected to decrease significantly even after the end of the pandemic [41,55]. Accordingly, better products and improved service are essential for strengthening competitiveness. We believe that the results of this study have provided important insights to achieve this goal. Regarding the current study, we suggest policy proposals to develop the food and food service industry. First, compared to 2019, keywords such as “discount,” “foundation,” and “Delivery App.” became strongly influential and valuable in 2021, and these may be applied to post-COVID-19 dining out trend analyses. Moreover, developers and marketing personnel should pay attention to the trends highlighted by the keywords “eating alone,” “cost-effectiveness,” “late-night meals,” “meal-kit,” and “premium” in product development and promotion. After the pandemic emerged, customers showed great interest in home meals, delivery food, and lunch boxes, and, as a result, delivery food increased. In this result, the main discourses common to consumers’ emotions regarding food delivery before and after the onset of COVID-19 are “enjoy,” “new,” and “satisfaction.” At the same time, the increase in negative keywords such as “fear,” “pain,” and “anger,” once COVID-19 emerged suggests that there is a lot of anxiety among consumers about food delivery due to the pandemic. Accordingly, food service companies need measures for hygiene and food quality. In addition, industry officials will need to pay attention and take action on important delivery-related issues that can cause negative emotions for consumers, such as delivery service problems and environmental pollution caused by waste. Lastly, as consumers have both positive and negative emotions about food delivery, emotional marketing should be applied so that they can feel confident in their decision to have food delivered.

## 6. Limitation and Future Studies

This study has several limitations. First, due to the scarcity of academic research and big data analyses of food delivery, a comparative analysis with previous studies could not be sufficiently carried out. Second, as this study was collected from two portal sites (i.e., Naver and Daum), it cannot be said that it reflects all opinions. Future research should include Instagram, Facebook, and Twitter to further supplement this work. Finally, it is difficult to generalize the situation in Korea’s food service market because countries have different containment and quarantine measures due to COVID-19. Therefore, even if the subject and research method of this study are applied to other countries, it is believed that different meaningful results can be drawn only in that country. Finally, as this study cannot accurately identify the actual factors that increased negative emotions after COVID-19, the interpretations related to negative emotions mentioned herein are estimates based on important issues related to food delivery services for consumers. Therefore, factors related to consumers’ negative emotions toward delivery services should be clarified through follow-up studies.

This study empirically presented consumers’ perceptions of food delivery service before and after the COVID-19 pandemic using big data. The results are expected to provide fundamental marketing data for researchers and industry officials seeking to research and develop delivery food products and services.

## Figures and Tables

**Figure 1 foods-11-03029-f001:**
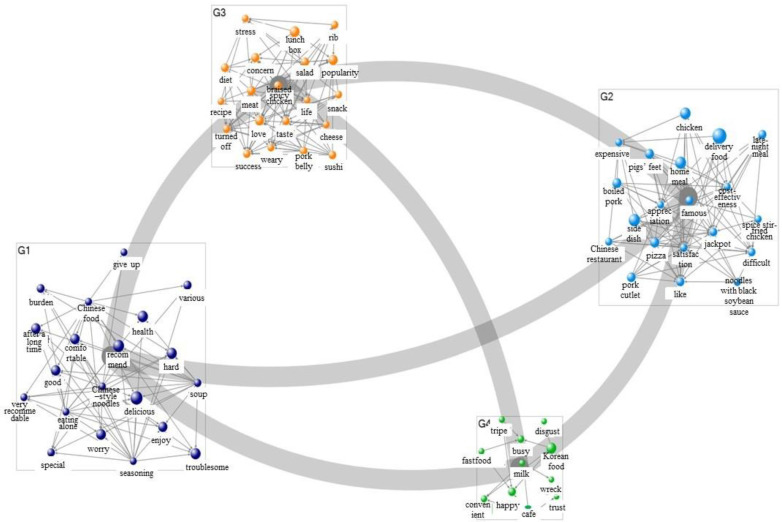
Sentimental network visualization of food delivery (2019).

**Figure 2 foods-11-03029-f002:**
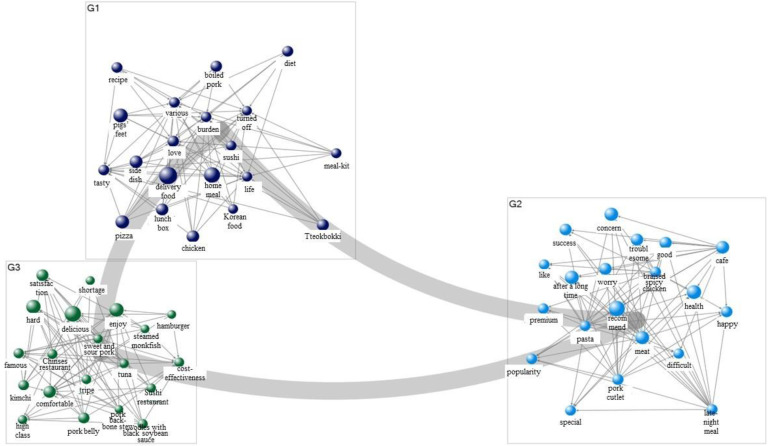
Sentimental network visualization of food delivery (2021).

**Figure 3 foods-11-03029-f003:**
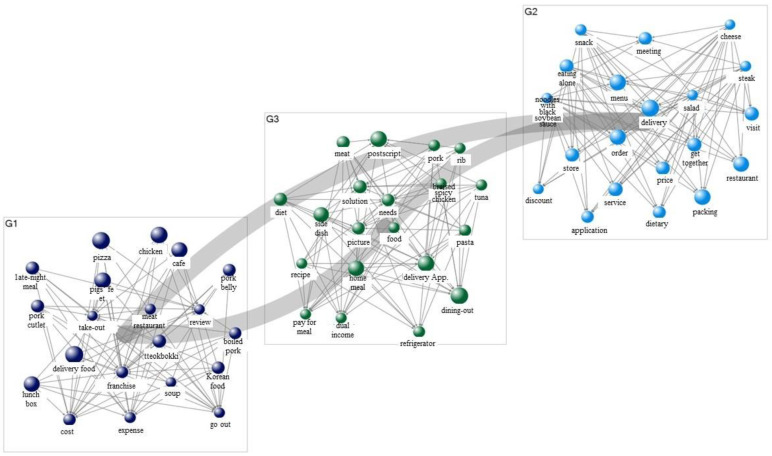
Demand network visualization of food delivery (2019).

**Figure 4 foods-11-03029-f004:**
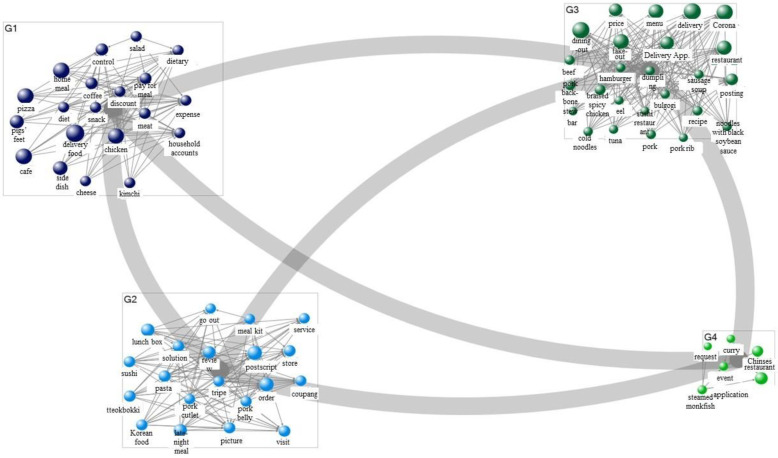
Demand network visualization of food delivery (2021).

**Table 1 foods-11-03029-t001:** Survey of collected data.

Data	Channel	Section	2019	2020
Dining out delivery	Naver	Blog	10,899	11,147
		Cafe	11,789	11,799
	Daum	Blog	9896	10,464
		Cafe	6560	5830

**Table 2 foods-11-03029-t002:** Narrative coding index.

Categories	2019	2021	Total
Food	320	302	622
Sentimental	113	107	220
Demand	227	255	482
Total	660	664	1324

**Table 3 foods-11-03029-t003:** Text mining of food delivery (2019).

Rank	Word	Frequency	TF-IDF	Rank	Word	Frequency	TF-IDF
1	dining-out	17,728	2460.8147	26	cooking	1049	3177.4030
2	delivery	15,650	7038.7212	27	recommend	1045	3117.8618
3	famous restaurant	5742	8919.3268	28	lunch box	897	3077.5019
4	delivery food	4869	6617.1284	29	application	877	2801.4071
5	foundation	2897	9425.7243	30	nowadays	858	2601.9370
6	dish	2632	5530.6962	31	make	828	2578.9686
7	family	2501	5336.8016	32	franchise	807	2581.2131
8	order	2170	4937.2079	33	health	758	2455.0274
9	delicious	2012	4666.4444	34	meal	756	2501.4223
10	pigs’ feet	1948	6052.3276	35	people	744	2379.7057
11	Delivery App.	1906	4881.6736	36	thought	720	2329.8407
12	chicken	1747	4782.8242	37	pork cutlet	717	2828.7611
13	menu	1700	4300.4077	38	about	716	2340.3397
14	side dish	1660	5183.3647	39	today	712	2346.9229
15	dinner	1513	3972.9890	40	place	707	2431.9524
16	take-out	1451	3878.7284	41	because	703	2270.7254
17	food	1373	3911.1928	42	mother	702	2403.5010
18	pizza	1315	4213.5058	43	lunch	681	2317.3142
19	children	1288	3736.0673	44	time	677	2231.5545
20	available	1260	3531.7626	45	tteokbokki	669	2435.0877
21	home meal	1257	3488.0423	46	industry	638	2278.9802
22	weekend	1238	3477.1712	47	sushi	637	2602.5469
23	restaurant	1216	3504.5128	48	everyday	634	2141.1387
24	market	1186	3462.7659	49	store	598	2047.4268
25	service	1103	3288.4038	50	pay for meal	580	2167.1809

**Table 4 foods-11-03029-t004:** Text mining of food delivery (2021).

Rank	Word	Frequency	TF-IDF	Rank	Word	Frequency	TF-IDF
1	dining-out	22,000	3715.7095	26	newspaper	1262	4277.7728
2	delivery	20,540	9001.6591	27	virus	1234	3516.0106
3	delivery food	6315	8625.8535	28	mask	1221	3482.9667
4	famous restaurant	5258	9725.3708	29	all day	1204	3455.3784
5	Corona	4311	7673.4030	30	wife	1198	3440.1589
6	dish	2875	6049.0050	31	shopping mall	1192	3429.9206
7	order	2623	6089.1432	32	simple	1188	3420.4091
8	application	2301	5952.7807	33	theater	1187	3423.5401
9	family	2279	5552.1404	34	time	1178	3518.1588
10	cooking	2139	5739.5662	35	weekend	1093	3494.9615
11	home meal	2053	5038.5068	36	pay for meal	1034	3522.4770
12	take-out	2015	5191.4708	37	method	1030	3190.3708
13	restaurant	1989	4958.4401	38	card	1020	3420.9380
14	menu	1848	4811.3006	39	government	1007	3465.5779
15	delicious	1757	4663.0262	40	chicken	1004	3403.2360
16	nowadays	1484	3978.9307	41	pizza	966	3579.3487
17	day	1473	3940.3103	42	pigs’ feet	935	3719.1911
18	meal	1419	3907.2500	43	people	935	3047.7806
19	because	1418	3854.8458	44	about	887	2908.9693
20	dinner	1412	3980.0125	45	support	881	3105.1834
21	discount	1405	4576.3637	46	review	871	2851.0424
22	foundation	1342	4528.4010	47	side meal	858	3209.3993
23	available	1339	3863.0711	48	kitchen	844	3588.7289
24	recommend	1331	3926.1170	49	lunch box	844	3279.8804
25	Delivery App.	1318	4019.1164	50	children	839	2827.6249

**Table 5 foods-11-03029-t005:** Sentimental network index of food delivery (2019).

Rank	Word	Degree Centrality	Betweenness Centrality	Closeness Centrality	Page Rank	Group	Category
1	delicious	69	779.6910	0.6635	0.0091	1	sentimental
2	recommend	65	599.7878	0.6389	0.0087	1	sentimental
3	health	53	525.5802	0.5750	0.0086	1	sentimental
4	troublesome	62	487.4306	0.6216	0.0085	1	sentimental
5	hard	59	454.9562	0.6053	0.0084	1	sentimental
6	worry	58	433.4342	0.6000	0.0084	1	sentimental
7	comfortable	57	408.1512	0.5897	0.0083	1	sentimental
8	after a long time	58	399.8882	0.6000	0.0082	1	sentimental
9	good	55	397.6608	0.5847	0.0082	1	sentimental
10	enjoy	55	367.5925	0.5798	0.0082	1	sentimental
11	various	45	243.2623	0.5391	0.0077	1	sentimental
12	burden	44	212.2904	0.5349	0.0076	1	sentimental
13	special	40	178.1011	0.5149	0.0075	1	sentimental
14	soup	27	144.9442	0.4742	0.0074	1	food
15	very recommendable	36	132.0809	0.5000	0.0073	1	sentimental
16	Chinese food	33	117.9320	0.4946	0.0072	1	food
17	Chinese-style noodles	35	116.4590	0.5018	0.0072	1	food
18	eating alone	31	114.6448	0.4876	0.0072	1	food
19	give up	32	109.8148	0.4859	0.0072	1	sentimental
20	seasoning	33	106.4018	0.4946	0.0071	1	food
21	delivery food	66	1130.4400	0.6479	0.0100	2	food
22	side dish	61	647.2332	0.6188	0.0088	2	food
23	home meal	59	612.8303	0.6079	0.0087	2	food
24	chicken	56	557.2379	0.5923	0.0087	2	food
25	pizza	52	348.0457	0.5726	0.0080	2	food
26	boiled pork	46	347.7718	0.5455	0.0080	2	food
27	pigs’ feet	51	322.2355	0.5679	0.0079	2	food
28	pork cutlet	40	304.2325	0.5208	0.0079	2	food
29	like	45	242.3503	0.5391	0.0077	2	sentimental
30	late-night meal	45	239.8520	0.5412	0.0077	2	food
31	famous	43	211.6894	0.5267	0.0077	2	sentimental
32	jackpot	40	200.1806	0.5188	0.0076	2	sentimental
33	difficult	40	181.7780	0.5149	0.0075	2	sentimental
34	satisfaction	37	154.8277	0.5074	0.0074	2	sentimental
35	noodles with black soybean sauce	33	151.5366	0.4946	0.0073	2	food
36	Chinese restaurant	37	139.4312	0.5092	0.0073	2	food
37	spice stir-fried chicken	33	121.1428	0.4946	0.0072	2	food
38	cost-effectiveness	34	115.8721	0.4929	0.0072	2	sentimental
39	appreciation	31	114.2020	0.4825	0.0072	2	sentimental
40	expensive	29	112.6933	0.4792	0.0072	2	sentimental
41	lunch box	52	421.8453	0.5726	0.0082	3	food
42	popularity	46	359.4550	0.5433	0.0082	3	sentimental
43	love	50	328.6673	0.5565	0.0081	3	sentimental
44	meat	47	271.6067	0.5498	0.0078	3	food
45	concern	45	264.6603	0.5391	0.0078	3	sentimental
46	salad	38	225.6139	0.5130	0.0076	3	food
47	diet	42	203.7948	0.5287	0.0076	3	food
48	braised spicy chicken with vegetable	42	199.8493	0.5287	0.0075	3	food
49	sushi	40	180.7834	0.5208	0.0075	3	food
50	pork belly	41	178.7188	0.5247	0.0074	3	food
51	success	38	168.5462	0.5111	0.0074	3	sentimental
52	tasty	39	163.3918	0.5149	0.0074	3	sentimental
53	rib	35	152.2859	0.5018	0.0073	3	food
54	turned off	34	133.5861	0.4929	0.0073	3	sentimental
55	snack	26	116.2942	0.4710	0.0072	3	food
56	weary	32	112.8708	0.4894	0.0072	3	sentimental
57	stress	29	95.4819	0.4759	0.0071	3	sentimental
58	recipe	30	90.0593	0.4842	0.0070	3	food
59	cheese	31	89.1592	0.4876	0.0070	3	food
60	life	28	82.1945	0.4726	0.0070	3	sentimental
61	Korean food	35	462.6095	0.5018	0.0085	4	food
62	happy	41	205.7489	0.5227	0.0076	4	sentimental
63	cafe	31	179.1014	0.4876	0.0075	4	food
64	busy	27	94.2732	0.4726	0.0071	4	sentimental
65	convenient	23	66.6541	0.4600	0.0069	4	sentimental
66	tripe	20	36.7552	0.4495	0.0067	4	food
67	fastfood	14	28.9633	0.4353	0.0066	4	food
68	milk	8	19.5987	0.4195	0.0066	4	food
69	wreck	3	2.4792	0.3988	0.0063	4	sentimental
70	trust	3	0.5122	0.3770	0.0062	4	sentimental
71	disgust	1	0.0000	0.3350	0.0062	4	sentimental

**Table 6 foods-11-03029-t006:** Sentimental network index of food delivery (2021).

Rank	Word	Degree Centrality	Betweenness Centrality	Closeness Centrality	Page Rank	Group	Category
1	delivery food	69	966.0612	0.6619	0.0094	1	food
2	home meal	64	723.2452	0.6318	0.0089	1	food
3	pig’s feet	56	541.5306	0.5890	0.0085	1	food
4	pizza	53	459.0090	0.5744	0.0083	1	food
5	side dish	49	382.4936	0.5560	0.0081	1	food
6	chicken	49	357.2233	0.5560	0.0080	1	food
7	lunch box	48	332.2937	0.5516	0.0079	1	food
8	tteokbokki	45	298.8448	0.5346	0.0078	1	food
9	love	44	256.6650	0.5346	0.0077	1	sentimental
10	boiled pork	42	256.4987	0.5265	0.0077	1	food
11	diet	41	223.5931	0.5226	0.0075	1	food
12	recipe	38	216.0452	0.5110	0.0076	1	food
13	tasty	42	214.0068	0.5226	0.0076	1	sentimental
14	various	40	195.8671	0.5187	0.0075	1	sentimental
15	burden	41	183.9379	0.5187	0.0074	1	sentimental
16	sushi	38	173.1633	0.5110	0.0074	1	food
17	meal-kit	39	161.6067	0.5148	0.0073	1	food
18	life	36	153.8467	0.5000	0.0073	1	sentimental
19	Korean food	33	134.7269	0.4929	0.0072	1	food
20	turned off	33	132.4612	0.4929	0.0072	1	sentimental
21	recommend	67	714.6530	0.6495	0.0090	2	sentimental
22	health	57	532.2749	0.5940	0.0086	2	sentimental
23	after a long time	58	477.2733	0.5991	0.0084	2	sentimental
24	concern	48	466.1040	0.5516	0.0083	2	sentimental
25	meat	53	460.5187	0.5744	0.0083	2	food
26	cafe	32	410.4856	0.4894	0.0081	2	food
27	troublesome	52	339.6440	0.5697	0.0079	2	sentimental
28	worry	52	329.4420	0.5697	0.0079	2	sentimental
29	good	51	320.8825	0.5650	0.0079	2	sentimental
30	success	43	306.5615	0.5305	0.0080	2	sentimental
31	happy	44	249.8971	0.5346	0.0077	2	sentimental
32	pork cutlet	41	236.5294	0.5226	0.0076	2	food
33	late-night meal	40	220.9788	0.5187	0.0075	2	food
34	braised spicy chicken with vegetable	40	210.8230	0.5187	0.0075	2	food
35	premium	31	209.8911	0.4826	0.0076	2	sentimental
36	special	38	200.9281	0.5110	0.0075	2	sentimental
37	popularity	42	197.6513	0.5265	0.0075	2	sentimental
38	pasta	39	189.1878	0.5148	0.0074	2	food
39	difficult	38	185.5199	0.5110	0.0074	2	sentimental
40	like	39	178.5465	0.5148	0.0074	2	sentimental
41	delicious	68	750.4686	0.6557	0.0090	3	sentimental
42	hard	60	533.0064	0.6096	0.0085	3	sentimental
43	enjoy	61	516.8716	0.6150	0.0085	3	sentimental
44	comfortable	53	344.5655	0.5744	0.0080	3	sentimental
45	satisfaction	49	303.8432	0.5516	0.0079	3	sentimental
46	pork belly	41	237.3934	0.5226	0.0076	3	food
47	Kim-chi	37	187.5835	0.5073	0.0074	3	food
48	famous	38	167.2956	0.5073	0.0074	3	sentimental
49	Chinese restaurant	35	147.6238	0.4964	0.0073	3	food
50	noodles with black soybean sauce	32	146.3486	0.4894	0.0073	3	food
51	cost-effectiveness	34	131.0722	0.4964	0.0072	3	sentimental
52	tripe	30	113.4425	0.4826	0.0071	3	food
53	high class	29	97.0923	0.4760	0.0071	3	sentimental
54	sushi restaurant	28	94.6259	0.4728	0.0070	3	food
55	sweet and sour pork	26	84.7977	0.4696	0.0070	3	food
53	shortage	25	76.9736	0.4633	0.0070	3	sentimental
57	tuna	27	75.5461	0.4728	0.0069	3	food
58	hamburger	21	64.4993	0.4542	0.0069	3	food
59	steamed monkfish	24	58.5209	0.4603	0.0068	3	food
60	pork back-bone stew	20	42.9154	0.4484	0.0067	3	food

**Table 7 foods-11-03029-t007:** Demand network index of food delivery (2019).

Rank	Word	Degree Centrality	Betweenness Centrality	Closeness Centrality	Page Rank	Group	Category
1	delivery food	67	572.8088	0.6495	0.0086	1	food
2	pizza	66	540.1352	0.6435	0.0085	1	food
3	chicken	63	492.0846	0.6261	0.0084	1	food
4	pigs’ feet	60	453.8836	0.6096	0.0083	1	food
5	café	60	424.7494	0.6096	0.0082	1	food
6	lunch box	60	422.3267	0.6096	0.0082	1	food
7	tteokbokki	49	259.5794	0.5560	0.0077	1	food
8	late-night meal	50	256.0452	0.5605	0.0077	1	food
9	pork cutlet	49	237.7509	0.5560	0.0076	1	food
10	Korean food	45	226.8937	0.5388	0.0076	1	food
11	pork belly	46	217.5030	0.5430	0.0076	1	food
12	cost	46	202.7864	0.5430	0.0075	1	demand
13	boiled pork	46	193.5608	0.5430	0.0075	1	food
14	franchise	45	183.0568	0.5388	0.0075	1	demand
15	expense	40	146.7131	0.5187	0.0073	1	demand
16	meat restaurant	36	131.8916	0.5036	0.0072	1	food
17	go out	39	124.9139	0.5148	0.0072	1	demand
18	review	38	114.0789	0.5110	0.0072	1	demand
19	soup	34	108.8485	0.4964	0.0071	1	food
20	take out	35	99.9595	0.5000	0.0071	1	demand
21	delivery	70	534.3241	0.6683	0.0086	2	demand
22	menu	67	450.0020	0.6495	0.0083	2	demand
23	packing	66	441.7215	0.6435	0.0083	2	demand
24	order	65	419.4226	0.6376	0.0083	2	demand
25	restaurant	64	401.8859	0.6318	0.0082	2	demand
26	service	57	315.4035	0.5940	0.0080	2	demand
27	get together	58	307.2211	0.5991	0.0079	2	demand
28	visit	57	302.9679	0.5940	0.0079	2	demand
29	price	55	287.5788	0.5840	0.0078	2	demand
30	eating alone	51	284.4426	0.5650	0.0078	2	food
31	store	56	277.8611	0.5890	0.0078	2	demand
32	meeting	54	255.1113	0.5792	0.0078	2	demand
33	dietary	48	230.8308	0.5516	0.0077	2	demand
34	application	49	203.3346	0.5560	0.0075	2	demand
35	snack	40	151.8706	0.5187	0.0073	2	food
36	steak	36	130.3094	0.5036	0.0072	2	food
37	noodles with black soybean sauce	36	123.2655	0.5036	0.0072	2	food
38	discount	38	118.4187	0.5110	0.0072	2	demand
39	salad	36	110.4815	0.5036	0.0071	2	food
40	cheese	37	103.9468	0.5073	0.0071	2	food
41	dining-out	70	534.3241	0.6683	0.0086	3	demand
42	review	67	456.3244	0.6495	0.0084	3	demand
43	Delivery App.	67	456.0833	0.6495	0.0084	3	demand
44	home meal	60	421.0417	0.6096	0.0082	3	food
45	side dish	57	385.6745	0.5940	0.0081	3	food
46	solution	54	264.1623	0.5792	0.0078	3	demand
47	diet	49	251.8059	0.5560	0.0077	3	food
48	meat	48	241.8891	0.5516	0.0076	3	food
49	picture	49	218.3996	0.5560	0.0076	3	demand
50	needs	48	214.8628	0.5516	0.0076	3	demand
51	pasta	42	181.6945	0.5265	0.0074	3	food
52	refrigerator	42	171.5882	0.5265	0.0074	3	demand
53	pork	39	161.2623	0.5148	0.0073	3	food
54	tuna	39	147.5488	0.5148	0.0073	3	food
55	rib	38	139.0739	0.5110	0.0073	3	food
56	food	35	135.3335	0.5000	0.0073	3	food
57	dual income	38	131.5006	0.5110	0.0072	3	demand
58	recipe	35	128.5534	0.5000	0.0072	3	food
59	braised spicy chicken with vegetable	38	127.8181	0.5110	0.0072	3	food
60	pay for meal	36	127.6059	0.5036	0.0072	3	demand

**Table 8 foods-11-03029-t008:** Demand network index of food delivery (2021).

Rank	Word	Degree Centrality	Betweenness Centrality	Closeness Centrality	Page Rank	Group	Category
1	delivery food	70	707.5897	0.6683	0.0090	1	food
2	café	63	543.0791	0.6261	0.0086	1	food
3	home meal	67	541.9647	0.6495	0.0085	1	food
4	pizza	66	531.5487	0.6435	0.0085	1	food
5	chicken	64	501.3976	0.6318	0.0085	1	food
6	side dish	58	386.0474	0.5991	0.0081	1	food
7	pigs’ feet	56	342.5173	0.5890	0.0080	1	food
8	control	53	245.2746	0.5744	0.0077	1	demand
9	coffee	46	221.7373	0.5430	0.0076	1	food
10	pay for meal	50	202.2793	0.5605	0.0076	1	demand
11	meat	46	185.0275	0.5430	0.0074	1	food
12	dietary	42	168.3363	0.5265	0.0074	1	demand
13	snack	43	165.7079	0.5305	0.0074	1	food
14	kimchi	44	162.0725	0.5346	0.0074	1	food
15	salad	45	156.7380	0.5388	0.0073	1	food
16	expense	43	155.4117	0.5305	0.0074	1	demand
17	household accounts	42	153.8515	0.5265	0.0074	1	demand
18	discount	44	149.6842	0.5346	0.0073	1	demand
19	diet	44	145.3571	0.5346	0.0073	1	food
20	cheese	40	143.5491	0.5187	0.0073	1	food
21	review	65	384.8104	0.6376	0.0082	2	demand
22	order	65	364.3533	0.6376	0.0081	2	demand
23	lunch box	55	297.3764	0.5840	0.0079	2	food
24	late-night meal	52	295.7392	0.5697	0.0079	2	food
25	review	55	248.6211	0.5840	0.0077	2	demand
26	pasta	47	199.0195	0.5472	0.0075	2	food
27	visit	50	197.2686	0.5605	0.0075	2	demand
28	store	50	195.4957	0.5605	0.0075	2	demand
29	tteokbokki	46	195.2880	0.5430	0.0075	2	food
30	picture	48	181.6599	0.5516	0.0075	2	demand
31	sushi	47	178.4888	0.5472	0.0074	2	food
32	solution	46	174.0137	0.5430	0.0074	2	demand
33	Korean food	44	171.8316	0.5346	0.0074	2	food
34	pork belly	44	159.9473	0.5346	0.0073	2	food
35	Coupang	44	151.4921	0.5346	0.0074	2	demand
36	service	44	146.5707	0.5346	0.0073	2	demand
37	tripe	39	139.7588	0.5148	0.0073	2	food
38	pork cutlet	40	133.7829	0.5187	0.0072	2	food
39	meal kit	36	131.5995	0.5036	0.0072	2	food
40	go out	42	130.4347	0.5265	0.0073	2	demand
41	delivery	70	601.6503	0.6683	0.0089	3	demand
42	dining-out	70	601.6503	0.6683	0.0089	3	demand
43	Corona	69	531.9884	0.6619	0.0086	3	demand
44	take-out	67	463.9959	0.6495	0.0084	3	demand
45	restaurant	65	413.1690	0.6376	0.0083	3	demand
46	menu	67	408.8245	0.6495	0.0082	3	demand
47	Delivery App.	63	332.5025	0.6261	0.0080	3	demand
48	price	59	292.2321	0.6043	0.0079	3	demand
49	posting	52	220.9994	0.5697	0.0076	3	demand
50	braised spicy chicken with vegetable	38	132.9270	0.5110	0.0073	3	food
51	recipe	34	91.1516	0.4964	0.0070	3	food
52	beef	34	89.9571	0.4964	0.0070	3	food
53	pork back-bone stew	31	83.7920	0.4860	0.0070	3	food
54	pork	33	78.7854	0.4929	0.0070	3	food
55	dumpling	30	70.0121	0.4826	0.0069	3	food
56	cold noodles	29	64.8984	0.4793	0.0069	3	food
57	noodles with black soybean sauce	27	64.3713	0.4728	0.0069	3	food
58	sausage soup	27	62.4482	0.4728	0.0068	3	food
59	hamburger	28	58.6885	0.4760	0.0068	3	food
60	bulgogi	28	55.1264	0.4760	0.0068	3	food
61	tuna	28	52.5336	0.4760	0.0068	3	food
62	eel	24	46.1091	0.4633	0.0067	3	food
63	pork rib	26	45.4708	0.4696	0.0067	3	food
64	bar	23	39.1435	0.4603	0.0067	3	food
65	sushi restaurant	25	38.7752	0.4664	0.0067	3	food
66	application	56	260.6288	0.5890	0.0078	4	demand
67	Chinese restaurant	42	164.4834	0.5265	0.0074	4	food
68	steamed monkfish	29	62.3813	0.4793	0.0069	4	food
69	event	26	48.7479	0.4696	0.0068	4	demand
70	curry	21	24.8492	0.4542	0.0066	4	food
71	request	10	4.6884	0.4238	0.0063	4	demand

**Table 9 foods-11-03029-t009:** Sentiment word frequency of food delivery.

	2019	2021	Increase or Decrease
Positive word	75.64	72.79	−2.85%
Negative word	24.36	27.21	+2.85%

**Table 10 foods-11-03029-t010:** Sentiment analysis of food delivery (2019).

	Frequency	Sentiment Intensity (%)	Frequency Percentage
good feeling	12030	62.41	61.37
joy	1208	5.69	6.16
interest	1589	8.02	8.11
Positive total	14,827	76.12	75.64
sadness	1319	5.84	6.73
disgust	2780	15.19	14.18
fear	287	1.09	1.46
pain	252	1.04	1.29
anger	98	0.49	0.50
fright	39	0.24	0.20
Negative total	4775	23.89	24.36
Total	19,602	19,602	100.00

**Table 11 foods-11-03029-t011:** Sentiment analysis of food delivery (2021).

	Frequency	Sentiment Intensity (%)	Frequency Percentage
good feeling	12877	63.82	60.50
joy	1653	7.93	7.77
interest	963	4.27	4.52
Positive total	15,493	76.02	72.79
sadness	1661	6.05	7.80
disgust	2640	12.19	12.40
fear	625	1.7	2.94
pain	569	2.57	2.67
anger	176	0.76	0.83
fright	121	0.71	0.57
Negative total	5792	23.98	27.21
Total	21,285	21,285	100.00

## Data Availability

The data used to support the findings of this study can be made available by the corresponding author upon request.

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
