# Peer review of "Analysis of Food Delivery Using Big Data: Comparative Study before and after COVID-19"

_foods, 2022, doi:10.3390/foods11193029_

Round 1

Reviewer 1 Report

First of all, I would like to congratulate the authors for their work.

They have conducted an interesting study to identify consumer perceptions of food delivery before and after the COVID-19 pandemic. The work is based on the collection and analysis of big data through social networks.

- Line 146. I think that the sentece "sentiment analysis was used" should be removed as it refers to one of the four methods mentioned just after (lines 147-148).

I think the obtention of some indexes should be better explained (TF-IDF, Degree centrality, closeness centrality, etc..). Specifically:

-Tables 3 and 4. How are the TF-IDF values obtained?

-Lines 243-244. In addition, the TF-IDF values of keywords such as such as “foundation,” “side dish,” “pizza,” “tteokbokki,” and “sushi” were substantially higher than those of other keywords. I do not understand this claim. For example, the TF-IDF value of “tteokbokki” is lower than that of many other words in Table 3 (it appears in position rank 45). The same for  keywords such as “famous restaurant,” “discount,”  “foundation,” “Baemin App.,” “pigs’ feet,” “side dish,” and “kitchen” referring to Table 4. Please, explain this better.

- How do you obtain the degree of centrality (Tables 5 and 6)?

- Figures 1, 2, 3 and 4 are not discernible. It is therefore difficult to understand the comments made about them.

- In Table 6 the most prominent values are highlighted in bold. However, in Table 5 this is not done. Pleas, homogenize.

- Lines 371-375. Check whether the following suggestions regarding the Tables quotes are appropriate:

In a comparison of 2019 and 2021 regarding food delivery, the number of positive keywords among sentiment-based words decreased by 2.85%, while the number of negative keywords increased by 2.85% (Table 9). In particular, sub-emotions under the positive category (e.g., good feelings, joy, interest) decreased in 2021 compared to 2019, while sub-emotions under the negative category (e.g., sadness, fear, pain) increased (Tables 10, 11).

- I believe that in Table 9, the +/- signs are swapped.

Author Response

Reviewer 1

We would like to thank the reviewer for thoughtful consideration of our manuscript. We have thoroughly studied reviewer’s comments and recommendations and found them very insightful and constructive; the reviewer provided valuable guidelines for improving the paper. The following paragraphs summarize our efforts in response to the reviewer’s comments and suggestions. Thank you very much for your kind notes!!

Comment#1: Line 146. I think that the sentece "sentiment analysis was used" should be removed as it refers to one of the four methods mentioned just after (lines 147-148). We appreciate the reviewer’s suggestions. We deleted it.

Comment#2: I think the obtention of some indexes should be better explained (TF-IDF, Degree centrality, closeness centrality, etc). We appreciate reviewers’ specific and helpful suggestion. We revised it as below.

(1) “Useful words were extracted, and major text-mining indexes, such as the frequency of occurrence and term frequency-inverse document frequency (TF-IDF; Term Frequency-Inverse Document Frequency), were calculated through text-mining.”

(2) “Additionally, the TF-IDF values of keywords such as “foundation,” “side dish,” “pizza,” “tteokbokki,” and “sushi” were substantially higher than those of other keywords, which signifies that these keywords have a very good scarcity value within documents retrieved with the keyword “food delivery” and are significant keywords although they do not appear frequently. Since TF-IDF plays an important role in a short-term trend analysis by factoring in both the frequency of words and the inter-document irregularity of the appearance of words, it could be assumed that these keywords acted as major factors in food delivery trends in 2019.”

Comment#3: Tables 3 and 4. How are the TF-IDF values obtained? → Thank you for your comments! It was derived through text mining technique.

Comment#4: Lines 243-244. In addition, the TF-IDF values of keywords such as such as “foundation,” “side dish,” “pizza,” “tteokbokki,” and “sushi” were substantially higher than those of other keywords. I do not understand this claim. For example, the TF-IDF value of “tteokbokki” is lower than that of many other words in Table 3 (it appears in position rank 45). The same for  keywords such as “famous restaurant,” “discount,”  “foundation,” “Baemin App.,” “pigs’ feet,” “side dish,” and “kitchen” referring to Table 4. Please, explain this better. → Thank you for your kind comment! It is interpreted meaningfully when a lower-ranked item has a higher TF-IDF value than a higher-ranked item.

Comment#5: How do you obtain the degree of centrality (Tables 5 and 6)? We appreciate reviewer’s kind advice. It is derived through Concor analysis.

Comment#6: Figures 1, 2, 3 and 4 are not discernible. It is therefore difficult to understand the comments made about them. → We greatly appreciate the time taken by the reviewer to point out the specific areas where improvements were needs. We trust that the editorial team will adjust the size of the picture.

Comment#7: In Table 6 the most prominent values are highlighted in bold. However, in Table 5 this is not done. Pleas, homogenize. Thank you for your comments. Table 5 has been corrected. Please see on Table 5.

Comment#8: Lines 371-375. Check whether the following suggestions regarding the Tables quotes are appropriate: In a comparison of 2019 and 2021 regarding food delivery, the number of positive keywords among sentiment-based words decreased by 2.85%, while the number of negative keywords increased by 2.85% (Table 9). In particular, sub-emotions under the positive category (e.g., good feelings, joy, interest) decreased in 2021 compared to 2019, while sub-emotions under the negative category (e.g., sadness, fear, pain) increased (Tables 10, 11). I believe that in Table 9, the +/- signs are swapped. We highly value reviewer’s comment on this issue. Table 9 has been corrected. Please see on Table 9.

Reviewer 2 Report

Dear authors,

My opinion is that the manuscript is well written in a clear way, clearly expresses its case and it is suitable for the journal audience. Analysis of the Food Delivery Using Big Data is a very important and current topic in the field and the approach used in the paper is in line with today's trends. The results are well analyzed and discussed and well linked to the findings of past studies.

Even though some limitations are presented, they could be more elaborated in the paper, especially in the context of world literature, not only in the context of Korea.

Author Response

Reviewer 2

Comment#1: My opinion is that the manuscript is well written in a clear way, clearly expresses its case and it is suitable for the journal audience. Analysis of the Food Delivery Using Big Data is a very important and current topic in the field and the approach used in the paper is in line with today's trends. The results are well analyzed and discussed and well linked to the findings of past studies. Even though some limitations are presented, they could be more elaborated in the paper, especially in the context of world literature, not only in the context of Korea. We would like to thank the reviewer for thoughtful consideration of our manuscript.  

Reviewer 3 Report

Dear Authors,

The topic is very interesting. The issues that are recommended to be paid attention to are attached. 

Analysis of the Food Delivery Using Big Data: Comparative Study before and after COVID-19

1.      In the abstract, the analyses should be mentioned, esp. the type of the analysis. Abstract as a whole should be revised and presented more systematically.

2.      The importance and the contribution of the study for the researchers and the practitioners should be given more deeply and clearly. It is very clear and understandable in the Discussion and Implications part.

3.      Figures are very difficult for the readers to understand, these might be given in one page seperate from the tables.

4.      There is something wrong with the flow of the following paragraph. Under the capture of "sentiment analysis", there is sentiment analysis.

P.3 L.146-148. "A sentiment analysis was used. For this purpose, text mining, a semantic network analysis, a CONCOR analysis, and a sentiment analysis were used."

P.4. L. 162-164. "Finally, a sentiment analysis, also called opinion mining, is a natural language-processing technique that allows researchers to automatically check the evaluations of and opinions about objects such as products and services".

5.      The portals' being the largest should be supported by objective measures and the function of those portals are not clearly explained for the ones who are unfamiliar to Korean media. Did the portals give permission to use their names?  

L. 194- 197. "...Naver and Daum. They are 194 the country’s major portal sites that gather news data from numerous media outlets and 195 show very active community and cafe activities; therefore, they contain the largest 196 amounts of relevant data in Korea."

6.      When  is it first outbreak in Korea? A normal calendar might not be suitable for this unless there is any declaration of the pandemic in Korea on 1st of Jan.

P.4. L. 200 "... a one-year period before the COVID-19 outbreak (from January 1 to December 31,..."

7.      Baemin App might be very well-known and popular in Korea but what is "Baemin App" for the others?

Author Response

Reviewer 3

We would like to thank the reviewer for thoughtful consideration of our manuscript. We have thoroughly studied reviewer’s comments and recommendations and found them very insightful and constructive; the reviewer provided valuable guidelines for improving the paper. The following paragraphs summarize our efforts in response to the reviewer’s comments and suggestions. Thank you very much for your kind notes!!

Comment#1: In the abstract, the analyses should be mentioned, esp. the type of the analysis. Abstract as a whole should be revised and presented more systematically. We highly value reviewer’s comment on this issue. We added it. “As analysis methods, results were derived through text mining analysis, Concor analysis, and sentiment analysis.”

Comment#2: The importance and the contribution of the study for the researchers and the practitioners should be given more deeply and clearly. It is very clear and understandable in the Discussion and Implications part. Thank you for your comments

Comment#3: Figures are very difficult for the readers to understand, these might be given in one page seperate from the tables. We appreciate reviewers’ specific and helpful suggestion. We trust that the editorial team will adjust the size of the picture.

Comment#4: There is something wrong with the flow of the following paragraph. Under the capture of "sentiment analysis", there is sentiment analysis. P.3 L.146-148. "A sentiment analysis was used. For this purpose, text mining, a semantic network analysis, a CONCOR analysis, and a sentiment analysis were used." / P.4. L. 162-164. "Finally, a sentiment analysis, also called opinion mining, is a natural language-processing technique that allows researchers to automatically check the evaluations of and opinions about objects such as products and services". We appreciate reviewer’s kind advice. Inaccurate and unnecessary sentences were deleted and revised. “This study attempted to identify consumers’ perceptions of food delivery before and after the onset of the COVID-19 pandemic by collecting and analyzing atypical data related to food delivery through social media. For this purpose, text mining, CONCOR analysis, and sentiment analysis were used. Text mining, a method of deriving meaningful keywords from unstructured data, is used for the purpose of deriving new information and knowledge [22,26]. In other words, it collects data related to natural language processing rather than structured data, establishes an analysis unit such as sentences or keywords in the text, and derives meaningful information based on the algorithm [22]. A CONCOR analysis is a method for explaining the meaning and characteristics of a cluster by classifying structural equivalence based on the correlation between keywords based on a semantic network analysis and then forming clusters with similar properties between keywords [27-32]. Finally, a sentiment analysis, also called opinion mining, is a natural language-processing technique that allows researchers to automatically check the evaluations of and opinions about objects such as products and services [33]. In general, a sentiment analysis is used to classify emotions expressed in texts or convert them into objective numerical information; in a narrow sense, it can be seen as classifying positive and negative emotions in text [34].

Comment#5: The portals' being the largest should be supported by objective measures and the function of those portals are not clearly explained for the ones who are unfamiliar to Korean media. Did the portals give permission to use their names? / L. 194- 197. "...Naver and Daum. They are the country’s major portal sites that gather news data from numerous media outlets and show very active community and cafe activities; therefore, they contain the largest amounts of relevant data in Korea." Thank you for your comments. Naver is Korea's No. 1 portal site, and it has been confirmed that no separate permission is required for the name of Naver to be exposed in manuscripts.

Comment#6: When is it first outbreak in Korea? A normal calendar might not be suitable for this unless there is any declaration of the pandemic in Korea on 1st of Jan. P.4. L. 200 "... a one-year period before the COVID-19 outbreak (from January 1 to December 31" We appreciate reviewers’ specific and helpful suggestion. In the case of Korea, the first case of Corona appeared on January 20, 2020, so it is reasonable to view it as of January.

Comment#7: Baemin App might be very well-known and popular in Korea but what is "Baemin App" for the others? We appreciate the reviewer’s suggestions. It has been revised to mean a “delivery app”. All manuscripts, figures, and tables have been revised.